# A Care Robot with Ethical Sensing System for Older Adults at Home

**DOI:** 10.3390/s22197515

**Published:** 2022-10-03

**Authors:** Jong-Wook Kim, Young-Lim Choi, Sang-Hyun Jeong, Jeonghye Han

**Affiliations:** 1Department Electronics Engineering, Dong-A University, Busan 49315, Korea; 2Department Computer Education, Cheongju National University of Education, Cheongju 28690, Korea

**Keywords:** robot ethics, care robot, frail older adult, human-centered artificial intelligence, human pose

## Abstract

Many studies have explored emotional and mental services that robots can provide for older adults, such as offering them daily conversation, news, music, or health information. However, the ethical issues raised by using sensors for frail older adults to monitor their daily movements or their medication intake, for instance, are still being discussed. In this study, we develop an older adult-guided, caregiver-monitored robot, Dori, which can detect and recognize movement by sensing human poses in accordance with two factors from the human-centered artificial intelligence (HCAI) framework. To design the care robot’s services based on sensing movement during daily activities, we conducted focus group interviews with two groups—caregivers and medical staff—on the topic of care robot services not for patients but for prefrail and frail elderly individuals living at home. Based on their responses, we derived the focal service areas of cognitive support, emotional support, physical activity support, medication management, and caregiver management. We also found the two groups differed in their ethical judgments in the areas of dignity, autonomy, controllability, and privacy for services utilizing sensing by care robots. Therefore, the pose recognition technology adopted in the present work uses only joint coordinate information extracted from camera images and thus is advantageous for protecting human dignity and personal information.

## 1. Introduction

Functional evaluations and health indicators for older adults are mainly constructed with consideration of the body (e.g., sight, hearing, activities of daily living [ADL]), cognition (e.g., cognitive decline, dementia), psychology (e.g., depression, solitude), and social environment (e.g., residential). Considering these various measures and needs, the potential of elder care robots in the home is significant. McGinn et al. [1] find socially assistive robots (SARs) can improve the working environment of caregivers while empowering older adults with independence and social connections. Grube et al. [2] state that health care, nursing, and community care are important factors to consider in a public health system for older adults, and that health care at home is just as important as nursing homes and nursing hospitals.

In this paper, the main target users of care robots are frail older adults who are not in good health but do not need to be hospitalized or live in long-term care facilities. Oh et al. [3] classify the health conditions of older adults as normal, prefrail, frail, or indicating a disorder. There is not a clearly defined standard for ‘frail’, so in the current study, we define prefrail and frail older adults as individuals who are living at home and undergoing outpatient treatment or exhibiting physical weakness and/or cognitive decline associated with aging. Older adults in the prefrail or frail stages need regular hospital checkups and may need medication management because of memory loss or mild dementia. If the older adult lives alone, an important service would be contacting their family or guardian in case of an urgent problem caused by, for example, falling or poor medication management.

Recently, memory training, medication guidance, and emotional exchange services have been developed for smart speakers and social robots with AI technology, as well as services based on the seven ADLs (i.e., dressing, washing, bathing, eating, mobility, transferring, toileting) and ten IADLs (instrumental activities of daily living; i.e., grooming, housekeeping, food preparation, laundry, going out short distances, managing transportation, shopping, handling finances, using a telephone, taking medication). However, as Gallagher et al. [4] report, robots are not yet ready to provide sophisticated services, but there will likely be improvements in the near future to care robots used in the home rather than to medical or nursing robots, which would require revision of medical law. Therefore, in this study, we develop an older adult-guided and caregiver-monitored robot using sensors and machine learning (ML) for use at home in the near future.

The following section reviews related studies on care robots and the human-centered artificial intelligence (HCAI) framework. Section 3 presents an ethical design for the sensing services of care robots by comparing the perceptions of dignity, autonomy, controllability, and privacy from two focus groups (caregivers and medical staff). Section 4 presents the development of the care robot Dori, which utilizes human pose recognition, and our consideration of the basic dignity and privacy of its frail older adult users as well as the values of their caregivers.

## 2. Related Work

In the case of care facility robots, research and development have mainly focused on patient functions that require physical labor, such as moving, eating, changing position, and going to the toilet. Care facility robots can be found in hospitals and nursing homes, such as Robear, which lifts patients from their bed to their wheelchair, and the Curaco Smart Bidet, an automatic toileting aid system. There are medical robots such as Moxi, which delivers medicines and diagnostic samples and connects patients with doctors through video calls. In Europe, in the Active and Assisted Living Programme (AAL), care robots support older adults in independent living at home and at nursing institutions. There are also elder care robots in Japan, such as Paro, which is in the form of a pet; Rudy, which is humanoid; Pepper, which provides recreation, medication management, exercise demonstrations, memory promotion activities, and family video call functions; Unazuki Kabochan, which provides conversation services for emotional support; and Sota, which controls home appliances and offers health advice.

According to Park et al. [5], care robot application services can involve greeting people, taking vital signs, playing music or fairy tales, brain fitness exercises, and photos and video for older adults and nursing staff. They found the most popular with older adults were music and photo services. Coco et al. [6] included heavy/light household work; moving older persons; assisting in washing, dressing, and using the bathroom; relieving anxiety and loneliness; and reminders for medication and scheduled meetings as services offered by care robots for workers in home care facilities. Unbehaun et al. [7] and Carros et al. [8] showed music (instrumental) and quiz applications provided by Pepper supported physical and cognitive activities for older patients and their caregivers. Pepper trained patients to follow its movements to music, hosted a quiz show, and played a team game with them.

Tuuli and Jaana [9] distinguished between robots with an effective design (e.g., patient-lifting robots) and robots with an affective design (emotional communication robots). To support affective design, developers should attend to the ethical problems that may occur with care robots and support the values and norms shared by caregivers. Sharkey and Sharkey [10] pointed out that the use of care robots could cause a potential reduction in face-to-face contact and increase objectification, loss of control, loss of privacy, loss of personal freedom, deception, infantilization, and situations in which people are unable to easily control robots (e.g., due to unfamiliar interfaces, lack of settings menu). They emphasized that care robots for older adults should aim to increase social contact, improve psychological well-being, and protect from threats. Berridge [11] argued that manual remote monitoring devices such as care robots and webcams can efficiently and remotely manage risks faced by older adult users by tracking, analyzing, and interpreting their location, activity, and movement, but these services can also cause problems. Older users can become extremely wary of their actions because sensor-based monitoring tracks changes in behavior or movement, and they can also be pressured into being monitored targets due to the patriarchal decision-making process. Berridge [12] also highlighted financial, social, and ethical cost-benefit issues, in areas such as privacy, autonomy, and prior consent, that are raised by manual remote monitoring devices, and the fact that less human contact can lead to isolation. Pullman [13] divided dignity into “basic dignity”, which is stable and sustainable, and “personal dignity”, which entails individual autonomy. Caregivers must attend to the preservation and enhancement of the basic and personal dignity of those entrusted to their care.

As passive remote monitoring technology expands in home- and community-based services, supporting informed decision-making, consent processes, and regulations for ethical and appropriate use is important [12]. The Akira AI Platform presents nine principles (in the areas of social welfare, unfair bias, privacy and security, trust and safety, transparency, controllability, value adjustment, responsibility, and human dignity) that provide a framework to help engineers be responsible in designing, developing, and maintaining systems for ethical AI in health care [14]. Additionally, Carros et al. [8] finds that in nursing homes, including caregivers in course setting and development strengthens participants’ trust in research and robots. Similarly, Gallagher [15] argues that two considerations should be taken into account in nursing practice—relationships with others (dignity) and self-relationship (self-esteem)—because undermining the nurses’ self-esteem undermines their ability to respect the dignity of patients, families of patients, and colleagues.

Shneiderman [16] proposes that HCAI be considered in life-critical systems in areas such as health care so they may be designed to be reliable, safe, and trustworthy (RST) to help facilitate human self-efficacy, mastery, and creativity. To illustrate how these concepts can manifest in the design of life-critical systems, he presented patient-controlled analgesia (PCA) devices, which distribute pain control drugs, on two axes of HCAI reflecting human control and computer automation. The resulting four PCA designs depend on different levels of patient control and automation: (1) morphine drip bag: regularly delivers fixed amounts of pain control medication, (2) automatic dispenser: provides automatic dose control based on patient activity and body signal sensor data, (3) patient-controlled device: patient controls the medication but the total amount is limited, (4) patient-guided, clinician-monitored device: RST design allows additional drugs but uses sensors and machine learning to select appropriate doses and prevent overdoses according to patient and disease variables, handles power or other errors, and includes hospital control center connections. In addition, Shneiderman [17] argues the design should take into account design goals, individual goals, and human values on top of the HCAI framework.

Therefore, the next section aims to design a care robot and accompanying services for frail older adults at home based on the fourth PCA device [16] by considering the older adults’ dignity, caregivers’ values, and design goals.

## 3. Designing of a Care Robot with Ethical Sensing of Human Poses

The AI research community is moving toward HCAI to calm fears of out-of-control robots, clarify responsibility for failures, and reduce biased decisions that lead to unfair treatment of minorities. In this section, we design a care robot and services based on both human mastery and RST in the HCAI framework, considering design aspirations (of caregivers, nurses, and doctors), individual goals (medication instruction; healthy cognitive, emotional, and physical abilities), and human values (dignity, control, autonomy). We assume that for the development of this care robot, the individual goals of older adults will be the same as their families, and thus matters of shared decision-making or intervention are not considered in this study.

### 3.1. Participants and Method

Caregivers, nurses, and clinicians were selected as a target audience for the design and the values and norms derived from their responses were used to inform the development of our care robot technology. Focus group interviews (FGIs) were conducted with a total of 47 people including 25 nurses and medical workers (5 males, 20 females) with an average experience of 13.81 years and 22 nursing care workers (2 males, 20 females) with an average experience of 7.17 years, to derive robot appearance, material, and care services, and to reflect user experience with the derived services in terms of human values and robot-based scenarios.

Section 3.2 examines the preferences of the caregiver group and the medical staff group for care robot services from previous studies. The services that are most preferred by both groups in the care of frail older adults will be selected for our robot. Section 3.3 compares the ethical judgments of the two groups on issues of medication management and the sensing process, particularly those related to the dignity and privacy of frail older adults and reflects them in the design of the care robot’s service architecture.

### 3.2. Selection of Services Based on RST

First, we explained the existing care robot services (mentioned in the studies in Section 2) to the participants using photos and videos. Then we introduced Dori, the robot we developed to support frail older adults in the home. Afterward, the participants individually responded to a mobile survey, selecting the services they thought Dori should provide as a care robot. The main services selected were cognitive activities, physical activities, and medication management. Next, we conducted an FGI with each group for further guidance on care robot services and caregiver management. Among the potential care services, video connection with children was also suggested with high frequency, but this service was not included because it can be conducted via voice or video call and does not raise ethical issues. Based on the results of the individual surveys and FGIs, six services were derived as follows:**Service introduction**: provide service items, basic settings, and usage guidelines**Cognitive activities**: provide quiz games based on calculation, word memory, etc., for cognitive enhancement**Emotional activities**: offer activities involving questions about user’s childhood or photos or singing along with familiar songs from their memory to counter loneliness and promote positive emotions**Physical activities**: offer games that detect movement of user’s body (arms and legs; e.g., the red light, green light game), or detect eating, falling, sleeping, etc.**Medication instruction**: detect, recognize, and approach user, tell them it is time to take their medication, and present the medication**Caregiver management**: keep nurse’s notes for regular caregiver to review health care information, check living environment, provide summary of health information check, and alert guardians if necessary

Services with an effective design include search and medication instruction, cognitive and physical activities, and caregiver management, and services with an affective design include emotional support. Cognitive activities, physical activities, and medication instruction services are designed around work processes, as shown in Figure 1, but in the case of emotional support activities, the values and norms shared by the group of care workers are considered important.

The satisfaction levels (1 = Very dissatisfied, 7 = Very satisfied) of the two groups with the five derived services were significantly more than four points, as shown in Table 1, and the two groups were equally satisfied.

### 3.3. Service Design of Ethical Sensing

The human values considered in the design of the care robot’s monitoring and managing services were dignity, protection of privacy, and control over the robot. To execute the physical activity services offered by the care robot, ADLs (e.g., eating, sleeping, toileting, mobility) and medication management require sensor processing to detect human poses. A record of meals, sleep, bowel movements, and falls would be especially useful information in the care of frail older adults who may also have memory problems. However, while using image sensing to detect falling or bathroom use may be directly related to life maintenance, at the same time, it can infringe on dignity. Similar to Shneiderman’s patient-controlled analgesia device that conformed to the HCAI framework [16], a care robot should adopt a patient-guided and clinician-monitored design. However, how should ethical standards involving ADLs and medication management for frail older adults be set? Do the two focus groups have different criteria for assessing dilemmas associated with dignity and privacy? Which group—caregivers or medical staff—best corresponds to the role of the clinician in regard to care services in the home?

We asked each group of participants to respond to two ethical dilemmas involving the robot’s monitoring and managing services using their mobile phones. In a second set of mobile surveys and FGIs, we asked the same participants to respond on the dilemmas, presented as follows:**Dilemma 1—Physical activities:** Mr. A has nearly fallen twice recently. Therefore, he leaves the bathroom door open and allows the care robot to monitor him. However, Mr. A does not want to let the care robot monitor him while he takes off his clothes in the bathroom. Should he stop or continue the care robot’s body image sensing when he is in the bathroom?**Dilemma 2—Medication management**: Mr. K has been suffering from excessive sleepiness whenever he takes his lunchtime medication. Due to this, he does not want to take his lunchtime medication today and ignores the instructions of the care robot. The care robot continues to hover around Mr. K when he does not take the medicine and to instruct him to take it. Should he stop or continue the care robot’s medication management?

Table 2 shows the results of an analysis of the two groups’ ethical judgments on human dignity, control and autonomy, and personal information issues related to the filming, recording, and transmitting of the user’s physical condition in Dilemma 1. First, when asked whether observation and filming by care robots undermines human dignity, the caregiver group judged the actions as moral behaviors necessary for treatment and care, with average scores of 6.05 and 5.23, respectively. The medical staff group judged the observational behavior with an average of 4.88, meaning they tended to view the action as treatment rather than damaging to dignity. However, they judged the robot’s filming behavior with an average of 3.96, viewing it as having the potential to undermine dignity. Overall, the medical staff group showed a neutral tendency to leave ethical judgment standards to the older adults and their families in regard to filming, although observation was possible.

When asked about controllability, whether older adults or their families should be authorized to change the settings of various services such as observation and photography by care robots, both groups agreed, with an average score of 4.86 for the caregiver group and 5.08 for the medical staff group. This is in line with the previous studies mentioned in Section 2 that suggest older adults should understand the functions, risks, and benefits of the monitoring technology in care robots and be provided with consent procedures so they can make decisions autonomously. In the event that the care robot is ordered to stop filming by the older individual, the caregiver group judged more strongly that the robot should not stop immediately, with an average score of 3.82, while the medical staff group indicated the robot should stop immediately, with an average of 5.28. The difference in opinions between the two groups was significant and suggests that caregivers strictly view care functions such as observing and photographing as part of their job, similar to administering medication or keeping status records. Finally, the two groups responded to the question of whether tracking falls and bowel movements are unethical due to the gathering of personal information. The average scores of both groups indicated that fall detection and bowel records were important to older adults’ health. Notably, the caregiver group judged these actions were not immoral.

In the case of the caregiver group, the results showed that the responsibility for actions to take care of the frail older adults for the service of the care robot has a strong ethical judgement. To satisfy the caregiver’s priorities while respecting the older individual, Dilemma 1 may be solved not by processing images of the user but by detecting their body’s joint coordinates using Google’s MediaPipe Pose and calculating merely the joint movements from camera images, further explained in the next section.

Although there was no difference in ethical decision-making between the two groups in their responses to the second dilemma, as shown in Table 3, the caregiver group generally agreed with the active medication instruction compared to the medical staff group. Both groups opposed stopping the medication immediately when the patient refused to take it, and many said it would be better if the care robot offered the medication at least three times. If the medication is ultimately refused, they said the care robot should keep a record of this situation and notify medical staff or the user’s guardian.

Since the purpose of this robot is to provide supportive care services, partially taking over the role of caregivers, its ethical settings were adopted based on the ethical judgments of the caregiver group. Thus, if the user refuses to take their medication, we designed the care robot to stop its recommendation after three prompts and to send a record of their medication intake to family and medical staff.

## 4. Development

### 4.1. Robot Development Environment

First, regarding the appearance of the elder care robot, we chose the concept of a teddy bear, common in homes, because older people have historically tended to prefer furry, compact doll-type robots that exhibit emotional stability similar to PARO-type robots. Figure 2 shows the appearance and configuration of our developed care robot, Dori, which is equipped with a microphone, speakers, sensors, and a medication dispenser. Dori features a 2D 360° LiDAR, Intel RealSense camera, and touch LCD on the Turtlebot 3 Waffle Pi robot platform by Robotis.

The LiDAR performs simultaneous localization and mapping [18] and navigation [19] required for autonomous driving. The camera conducts facial recognition, posture recognition, and object recognition required for user identification and service performance. The Intel NUC 11 Performance Kit is adopted as the main controller. The entire system is based on the Robot Operating System (ROS) [20] and uses the open-source control module OpenCR to control the wheels’ motors. In addition, Dori’s software framework is developed by connecting and integrating the cognitive agent architecture Soar [21], chatbot engine ChatScript [22], DB management system SQLite [23], and cloud system and open-source packages for voice and image recognition processes.

### 4.2. Software Architecture

Dori’s software architecture consists of five units: system input/output, voice and image recognition, chatbot processing, service setting, and service content.

**System input/output unit**: modules that handle robot drive and input data from the robot’s microphone, camera, and various sensors;**Voice and image recognition unit**: module that processes STT (Speech-to-Text) and TTS (Text-to-Speech) using Google Assistant APIs [24] through microphones and speakers, and translates English to Korean through Papago NMT (Neural Machine Translation) [25]; face recognition uses the camera and OpenFace [26], pose recognition uses MediaPipe Pose [27], and object recognition uses SSD-MobileNet [28];**Chatbot processing unit**: module using ChatScript for part-of-speech tagging, parsing, and parse analysis according to service scenarios;**Service setting unit**: elderly-controlled control module that allows the halting of care services containing personal information and that allows only video recognition processing when monitoring;**Service content**: information services (e.g., weather, news, time, schedule, reminders, phone numbers), care services (e.g., cognitive, emotional, physical, medication management, meal checks, fall detection), and reporting services (interactions with older users and health care information); care services can be stopped in the service setting unit; reporting service is monitored by caregivers;

The overall structure of Dori’s software framework is illustrated in Figure 3.

### 4.3. Care Services Based on HCAI and Human Values

Based on the results of the focus group interviews (Table 2 and Table 3), older adult users and their families were given autonomy and control over Dori’s care and reporting services, which can be set after listening to all of the service options. In addition, using the service setting unit, Dori can be set to record only the occurrence of events such as falls and medication intake by performing real-time pose recognition processing without photographing or storing camera image data (Figure 3).

MPP builds pipelines and processes cognitive data in the form of video using ML. MPP uses a BlazePose [29] approach for tracking human poses by inferring 33 landmarks. BlazePose is a lightweight convolutional neural network architecture for human pose estimation that is tailored for real-time inference. Each MPP landmark is represented by 2D joint coordinates normalized to image width and height within the range of [0.0, 1.0] for each RGB image frame. Among the extracted landmarks, we use 17 landmarks to estimate arbitrary poses and motions, whose indices are 0, 2, 5, 7, 8, 11, 12, 13, 14, 15, 16, 23, 24, 25, 26, 27, 28. The 2D joint pixel coordinates of MPP landmarks can be directly applied to physical activity function of Dori with the following steps:Step 1. Acquire images from camera with the image grabber module of Dori.Step 2. Extract 2D pixel coordinates of the 17 landmarks for the captured human body images of each frame by executing MPP.Step 3. Analyze the joint movement according to the change in normalized pixel coordinates by checking whether any of the current joint positions has moved above the normalized pixel distance threshold compared to the average position of the previous 4 frames.Step 4. If any joint of the frail older adult has changed during the red light-green light game or the whole body is moved as a result of Step 3, the information is sent to the Dori’s Care Service module shown in Figure 3.Step 5. If the current image frame is the last one or the termination condition is met, stop the motion detection process. Otherwise, go to Step 1.

Figure 4 shows that MPP successfully extracts the overall coordinates of 33 landmarks of each image for the motion of raising the arm upward and then lowering it to the side. In the figure, it can be seen that the joint coordinates of the upper body dynamically change, but the joint coordinates of the lower body are almost fixed. Figure 5 shows the corresponding normalized pixel distance profiles of the positions in shoulder, elbow, wrist, hip, knee, and ankle joints for this motion. Comparing the fluctuating profiles of the elbow and wrist joints with others, 0.05 is determined as a good criterion for distinguishing between active and static joints, which can be applied to detecting sudden movement of the older adult. When body movement is detected, the pose recognition function is activated.

For recognition of the older adult’s significant motions, the authors developed a human pose recognition package and implemented it in Dori, which uses a global optimization method, univariate Dynamic Encoding Algorithm for Searches (uDEAS) [30], to reconstruct the 2D joint coordinate information extracted from the MPP into a 3D human pose by adjusting the joint angle variables of the humanoid robot model [31]. In the current version, the humanoid model is updated by adding three lumbar joints, and the search principle of uDEAS has been modified to change the search order during optimization according to the effect of search variables on the cost function to accelerate optimization time.

Figure 6 shows the results of the MPP execution for the sitting and standing up motion images in the upper row and the corresponding 3D humanoid poses fitted to the 2D images by uDEAS in the lower row. Figure 7 shows the angular trajectories of the shoulder, elbow, hip, knee joints in the sagittal plane identified for this motion. In the figure, it is clear that the hip and knee joints reach about 70 degrees at frame 20, expressing a sitting motion with the humanoid model, and return to zero for a standing motion.

For validation of the pose recognition performance, we try to recognize a sudden falling motion as shown in Figure 8 and Figure 9. The images captured in Figure 8 and the poses of the humanoid simulation match well, and the estimated joint angle trajectories plotted in Figure 9 has apparently different feature compared to Figure 7. Specifically, the fact that the hip joint angle in the sagittal plane changes from 0 degree (standing) to −70 degrees (abnormal) means that the older adult is more likely to fall to the floor. As such, other poses can be recognized based on the representative joint angle profiles for Dory’s monitoring service.

Dori tracks and observes users in real time immediately after the monitoring service starts (Figure 10). The RGB images and depth values are received from the camera to determine the location of the user and the distance between the user and the robot. The direction the robot faces is continuously controlled so that the center point of the pelvis of the skeletal model acquired from the camera image is located at the center of the camera image frame using MediaPipe Pose. An appropriate distance between the pelvic coordinates and the robot is maintained using the depth information from the camera.

If the user expresses an intention to reject monitoring services, for example, by saying, “Do not follow” or “Turn off the camera”, Dori will attempt to reconfirm the user’s intention by explaining its justification and necessity. If the user says “no” again, Dori will respect the user’s intention and stop the camera from working. However, if Dori is performing a specific caregiver assistance function, even if the user expresses a denial of service, Dori is set to repeatedly confirm the user’s intention up to three times, based on the survey results in Table 3.

Figure 11 shows an example of the flow of a Dori’s chatbot conversation conducted with STT and TTS via Google Assistant API, a service scenario written in ChatScript, and the application of Papago NMT for language translation. The Korean sentences in Figure 11 are translations of the parenthetical English sentences, which are the responses that are actually coded into the service scenario in ChatScript. Since the present services use Korean, Papago NMT is necessary for respectful conversation or requests by Dori, as the Korean language has honorific expressions for the elderly or people of high status.

The sentences in angle brackets (<< >>) represent hardware modes and functions performed on Dori’s software framework, such as turning the camera on/off and human following/monitoring. These operations are harmonized with ROS message handling capabilities. Figure 12 illustrates the internal processes that are executed during conversations between Dori and the user.

## 5. Conclusions

As the elderly population rapidly increases, there is a growing demand for care services such as cognitive activities, emotional wellness activities, physical activities, and medication services for older adults in prefrail and frail stages, in addition to care services for transportation, eating, posture transformation, and bowel management for those in nursing homes or nursing hospitals. With the development of care robots and services, maintaining human dignity and caregiver values is an important ethical issue. Therefore, in this research, we designed Dori, an elderly-controlled, caregiver-monitored robot that fits within the HCAI framework for prefrail or frail older adults at home. We conducted focus group interviews with caregivers and medical staff regarding existing elder care services, and using their responses, we developed a program of care robot services including cognitive support, emotional support, physical activity support, medication management, and caregiver management. Equipped with a microphone, speakers, sensors, medication distributor, LiDAR, camera, touch LCD, comforting teddy bear appearance, and a module that can be controlled by the user, Dori is developed to attend to the basic dignity of older adults and the values of their caregivers.

The software architecture consists of five units (system input/output unit including autonomous driving, AI processing unit for voice and image recognition, chatbot processing unit, service setting unit, and service providing unit). In the service setting unit, users can enter their own settings and receive real-time video processing without having their video data stored separately through the service providing unit. If users reject a particular service, Dori does not immediately stop the service, but first explains the legitimacy and necessity of the service, confirming the user’s intention and attempting to fulfill the given task.

In developing our care robot prototype, we conducted a survey of indirect user experience for the application of robot-based services by providing service images to groups of caregivers and medical staff. Satisfaction with care robots and services, measured on a 7-point scale, was 5.8 (Table 1). Specifically, satisfaction for medication management services averaged 5.98, cognitive function services, 5.86, emotional function services, 5.96, physical function services, 5.57, and caregiver management services, 5.61, showing overall satisfaction. However, in the case of physical activity, the average was slightly lower than for the other services, as there were opinions such as “Needs more active activities rather than sitting down”, “The robot needs to demonstrate activities with its robotic body, instead of presenting videos on its chest monitor”, and “The height of the robot is somewhat small for older adults to do standing activities”.

According to a cloud letter analysis of the participants’ opinions, they considered the most important aspects of the care robot to be convenience and a practical appearance for older adults, followed by the practicality of the content implemented by the service. In addition, the size of the robot and the alert function (for accidents and medication management) were important, followed by the ability to immediately order a service to stop and error handling. Future research will be a field experiment in which Dori is actually operated as a long-term test service to older adults at home.

## Figures and Tables

**Figure 1 sensors-22-07515-f001:**
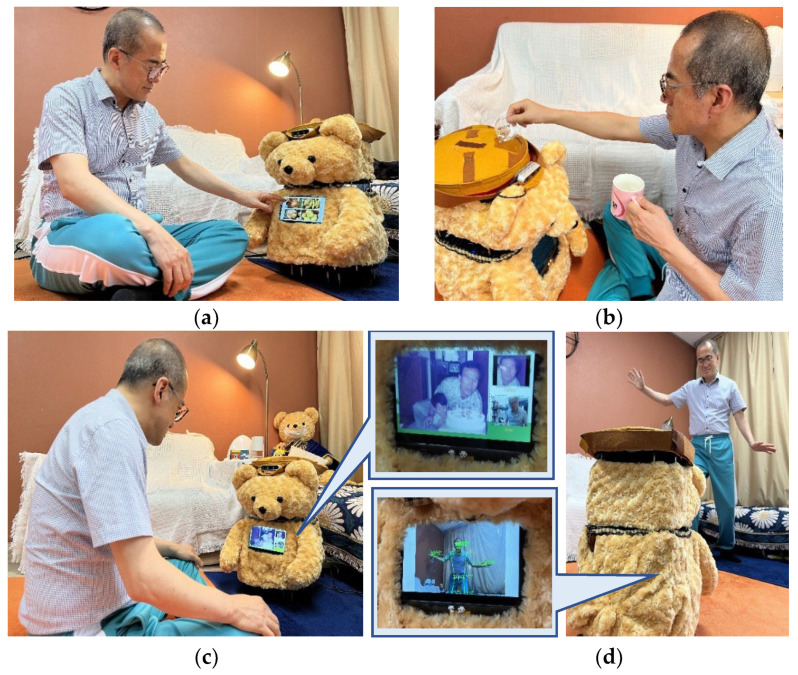
Scenes illustrating care robot services: (**a**) cognitive activity in which the user chooses a food item and then calculates the price of the food; (**b**) medication instruction and distribution of medication at a specific time; (**c**) emotional activity involving talking about childhood photos; (**d**) game involving use of body movement.

**Figure 2 sensors-22-07515-f002:**
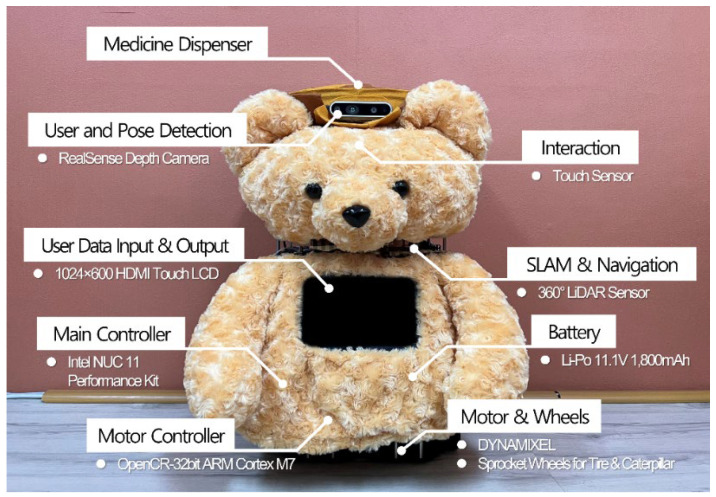
Hardware components of care robot DORI.

**Figure 3 sensors-22-07515-f003:**
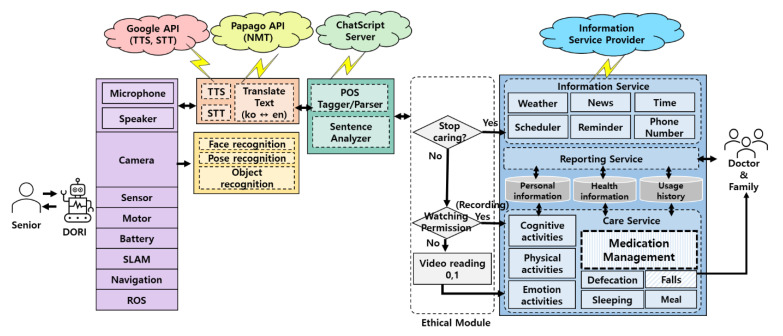
Software architecture of care robot Dori.

**Figure 4 sensors-22-07515-f004:**
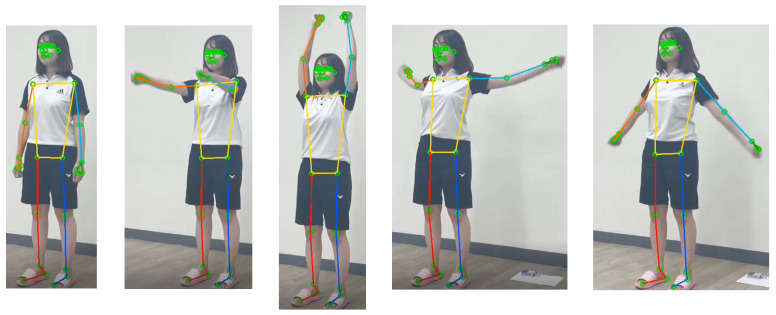
MPP execution results for the motion of raising the arm upward and then lowering it to the side.

**Figure 5 sensors-22-07515-f005:**
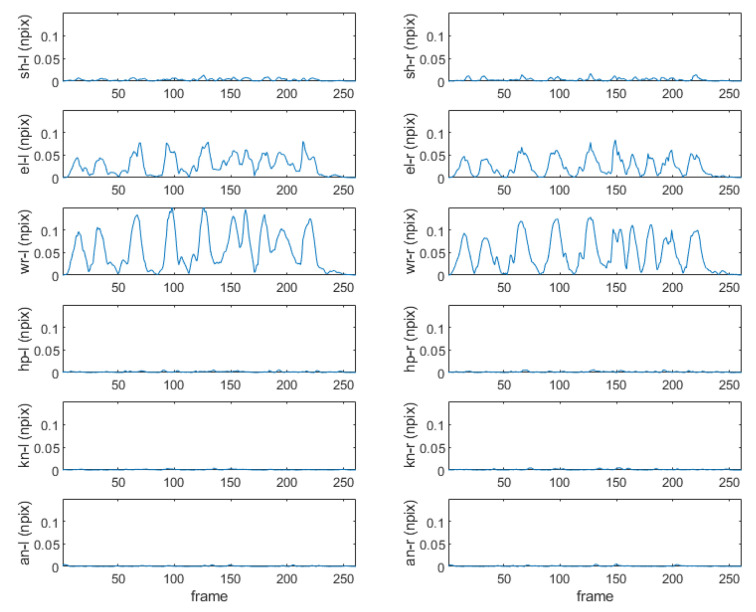
Change distance profiles of left (l)/right (r) shoulder (sh), elbow (el), wrist (wr), hip (hp), knee (kn), and ankle (an) joint landmark positions in normalized pixels (npix) for the motion in Figure 4.

**Figure 6 sensors-22-07515-f006:**
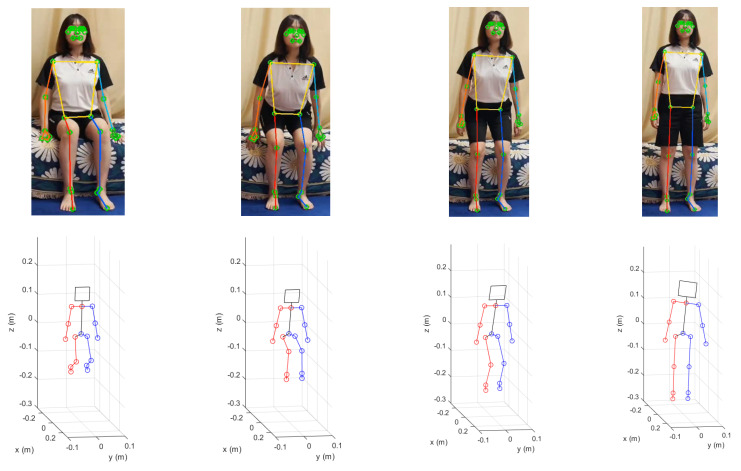
Results of MPP execution for the sitting and standing up motion images (upper row) and the corresponding humanoid poses reconstructed by the optimization method, uDEAS (lower row).

**Figure 7 sensors-22-07515-f007:**
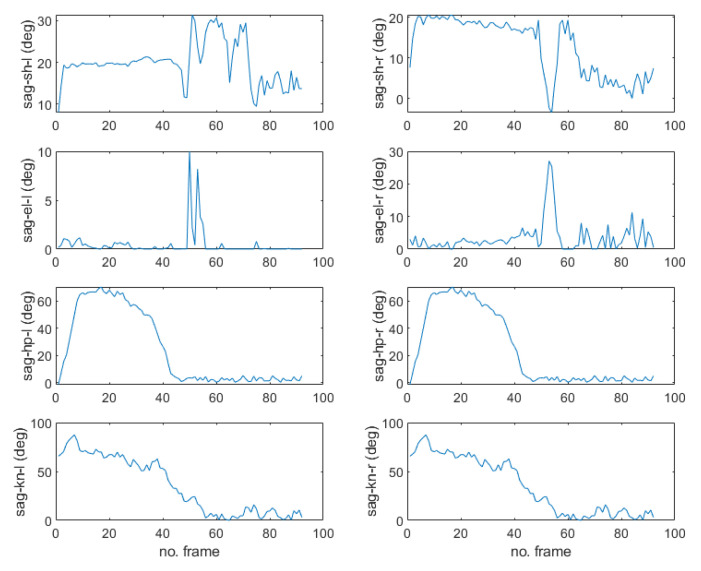
Angular trajectories (in degrees) of the shoulder, elbow, hip, and knee joints in the sagittal plane identified for the sitting and standing motion.

**Figure 8 sensors-22-07515-f008:**
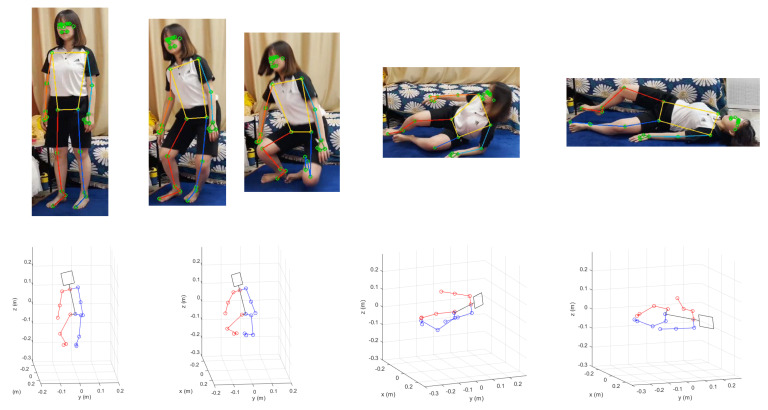
Results of MPP execution for a sudden falling motion images (upper row) and the corresponding humanoid poses reconstructed by uDEAS (lower row).

**Figure 9 sensors-22-07515-f009:**
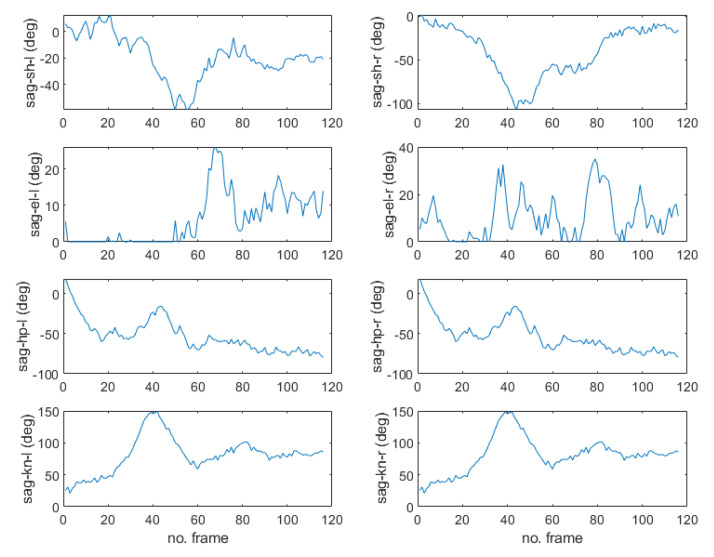
Angular trajectories (in degrees) of the shoulder, elbow, hip, knee joints in the sagittal plane identified for the sitting and standing motion.

**Figure 10 sensors-22-07515-f010:**
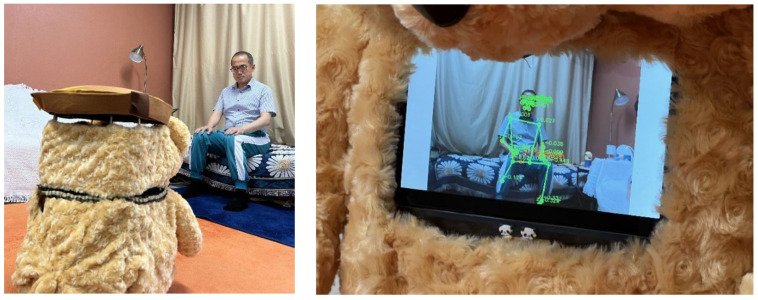
Left: Dori and the user under monitoring service. Right: Skeletal model is extracted from camera image using MediaPipe Pose.

**Figure 11 sensors-22-07515-f011:**
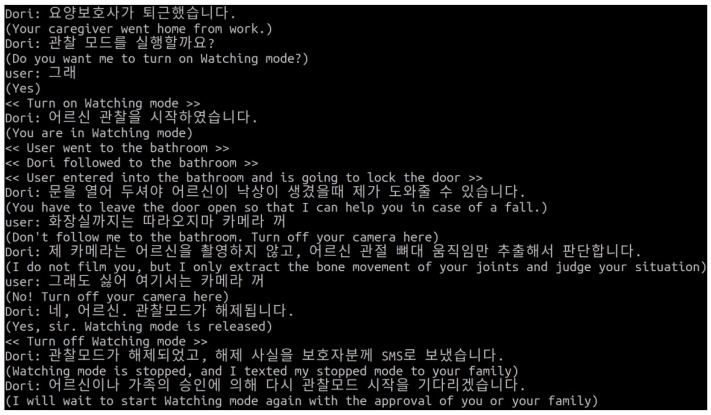
Example of a conversation between user and chatbot related to the second dilemma.

**Figure 12 sensors-22-07515-f012:**
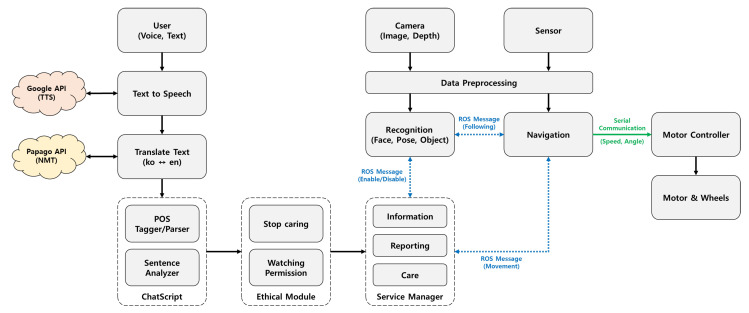
Flow diagram of internal processes executed during conversations between Dori and the user.

**Table 1 sensors-22-07515-t001:** Comparison of satisfaction with care robot services.

Care Robot Service	Average Satisfaction Score (STDDEV)		*T* Value	*p* Value
	Caregivers	Medical Staff		
Cognitive activity	6.05 (1.77)	5.64 (2.23)	0.96	0.344
Emotional activity	6.36 (1.78)	5.6 (3.04)	1.63	0.109
Physical activity	5.82 (2.69)	5.32 (2.62)	1.02	0.311
Medication instruction	6.05 (2.59)	6.04 (1.64)	0.01	0.990
Caregiver management	5.86 (2.94)	5.4 (2.32)	0.96	0.342

**Table 2 sensors-22-07515-t002:** Ethical judgment survey results of caregivers and medical staff on Dilemma 1.

Value	Ethical Judgment on Dilemma 1	Average (STDDEV)		*T* Value	*p* Value
		Caregivers	Medical Staff		
Dignity	Which of the two do you think the observation behavior of care robots is closest to? (1 = Impairment of dignity, 7 = Action for treatment)	6.05 (2.95)	4.88 (3.55)	2.16	0.036 **
	Since the care robot is not a person, do you think that filming by the care robot does not undermine the dignity of frail older adults? (1 = Impairment of dignity, 7 = Does not impair dignity with safety information)	5.23 (5.36)	3.96 (3.96)	1.97	0.054
Controllability	Do you think frail older adults or their guardians should be authorized to change ethical settings related to personal information, such as whether or not to film in sensitive situations? (1 = Do not allow changes by older adults or guardians, 7 = Allow changes by older adults or guardians)	4.86 (6.12)	5.08 (3.51)	−0.33	0.741
	If the older adult asks the care robot to stop filming, do you think the care robot should comply immediately? (1 = Should continue, 7 = Should stop immediately)	3.82 (5.33)	5.28 (2.76)	−2.46	0.018 **
Personal information	Do you think that the service of a care robot tracking and detecting falls is more important than the dignity or privacy of frail older adults? (1 = Privacy is more important, 7 = Fall detection is more important)	6.41 (1.70)	5.48 (2.81)	2.06	0.046 **
	Do you think that providing information about a frail older adult’s bowel movements and health status to family members is an infringement of their privacy? (1 = Serious privacy infringement, 7 = Noninfringement of privacy and more important to understand health status)	5.82 (2.97)	4.72 (3.48)	2.04	0.047 **

* *p* < 0.1. ** *p* < 0.05. *** *p* < 0.01.

**Table 3 sensors-22-07515-t003:** Ethical judgment survey results of caregivers and medical staff on Dilemma 2.

Ethical Judgment on Dilemma 2	Average (STDDEV)		*T* Value	*p* Value
	Care- Givers	Medical Staff		
Which of the two do you think best describes the continuous medication instruction by the care robot? (1 = Improper behavior ignoring Mr. K’s opinion, 7 = Very necessary action for health)	5.27 (3.56)	5.24 (2.74)	0.06	0.951
Do you think the care robot should require the frail older adult to take their medicine until they complete the prescribed dose? (1 = Strongly disagree, 7 = Strongly agree)	4.77 (4.99)	4.32 (3.74)	0.73	0.470
Do you think the care robot should stop giving instructions if the frail older adult does not respond or refuses to take the medicine even though they have been instructed about three times? (1 = Strongly disagree, 7 = Strongly agree)	4.82 (6.24)	4.8 (4.0)	0.03	0.979
Do you think the care robot should stop prompting the frail older adult immediately if they refuse to take their medication? (1 = Strongly disagree, 7 = Strongly agree)	3.09 (5.17)	3.2 (3.68)	−0.17	0.862

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
