# Peer review of "A Care Robot with Ethical Sensing System for Older Adults at Home"

_sensors, 2022, doi:10.3390/s22197515_

Round 1
Reviewer 1 Report
This paper proposes and designs a robot controlled by the elderly and monitored by the caregiver. The robot is equipped with cognitive support, emotional support, physical activity support, drug management, nursing staff management and other functions. This work is of great significance to the nursing service of the elderly, and has great practical value and social benefits.
The overall quality of the manuscript is fair, but there are some issues the authors should address and improve, in order to make their work suitable for possible publication in the journal. They are presented here:
1) English abbreviations appearing for the first time in the article should be spelled out in full, such as DORI.
2) The content of the first chapter is similar to that of the second chapter, which is redundant. It is recommended to combine the two chapters.
3) The English expression of the full text needs to be revised, and some expressions are confusing to the reviewers.
4) The average satisfaction score in Chapter 3 has only results and no basis. It is recommended to give further process descriptions.
Author Response
Respected Sir/Madam,
Thanks a lot for your valuable feedback. Please find below our point-to-point response to your comments.
Point 1: English abbreviations appearing for the first time in the article should be spelled out in full, such as DORI.
Response 1: Dori is the name of the robot and has no abbreviation. In the revised paper, DORI is modified to Dori. Others were also checked overall and corrected.
Point 2: The content of the first chapter is similar to that of the second chapter, which is redundant. It is recommended to combine the two chapters.
Response 2: We agree with the reviewer. We have deleted or moved the duplicate content of these two sections. In addition, the introduction mainly included the background, and the second section mainly summarized the research related to care robots. Most of Section1 and Section 2 have been revised.
Point 3: The English expression of the full text needs to be revised, and some expressions are confusing to the reviewers.
Response 3: Many English expressions have been modified, and this manuscript has been edited by a high-quality native speaker. In addition, confusing expressions have been reduced by unifying terms that are used repeatedly.
Point 4: The average satisfaction score in Chapter 3 has only results and no basis. It is recommended to give further process descriptions.
Response 4: Additional processes have been added to the service derivation process in Sections 3.2 and 3.3. In particular, Section 3.3 presents specific dilemmas, and also provided a summary of the various opinions obtained through FGI.
<Section 3.2>
First, we explained the existing care robot services (mentioned in the studies in Section 2) to the participants using photos and videos. Then we introduced Dori, the robot we developed to support frail older adults in the home. Afterward, the partici-pants individually responded to a mobile survey, selecting the services they thought Dori should provide as a care robot. The main services selected were cognitive activi-ties, physical activities, and medication management. Next, we conducted an FGI with each group for further guidance on care robot services and caregiver management. Among the potential care services, video connection with children was also suggested with high frequency, but this service was not included because it can be conducted via voice or video call and does not raise ethical issues.
<Section 3.3>
We asked each group of participants to respond to two ethical dilemmas involv-ing the robot’s monitoring and managing services using their mobile phones. In a sec-ond set of mobile surveys and FGIs, we asked the same participants to respond for the two groups were then conducted to collect their various opinions on the dilemmas, presented as follows:
Reviewer 2 Report
The title of the manuscript centers on the development of the robot, but the paper devotes a great deal of space to the development background, requirements, and related investigations, with little mention of the design, characteristics, and technical details of the robot development. This is not consistent with the title or the main idea, and it is recommended that the logical framework or the title be adjusted to make the paper consistent as a whole.
Due to the small amount of detail in the manuscript regarding the development of the design robot, we are unable to judge from the existing manuscript whether the work meets the acceptance criteria for this journal, so please add relevant work as appropriate so that readers can gain a deeper understanding of your work.
Author Response
Respected Sir/Madam,
Thanks a lot for your valuable feedback. Please find below our point-to-point response to your comments.
Point 1: The title of the manuscript centers on the development of the robot, but the paper devotes a great deal of space to the development background, requirements, and related investigations, with little mention of the design, characteristics, and technical details of the robot development. This is not consistent with the title or the main idea, and it is recommended that the logical framework or the title be adjusted to make the paper consistent as a whole.
Response 1: The title of the paper, the introduction, and the related research have been all revised. The title and clause structure and content have been also modified to ensure consistency throughout the manuscript structure.
Point 2: Due to the small amount of detail in the manuscript regarding the development of the design robot, we are unable to judge from the existing manuscript whether the work meets the acceptance criteria for this journal, so please add relevant work as appropriate so that readers can gain a deeper understanding of your work.
Response 2: We've modified a lot of the detail of the paper to link it to the topic of this journal. This manuscript includes ethical judgment and processing criteria for sensing data prior to the engineering method of collecting and processing sensing data.
In more detail, a simple introduction of MediaPipe Pose and explanation of the process for applying it to Dori’s physical activity function is added in Section 4.3. In addition, Figures 4 to 10 are added to describe and demonstrate our pose recognition technology that can be used in Dori’s monitoring service. We believe that our pose recognition technology combining MPP, a global optimization method, and humanoid model is up-to-date.
For more information, in Section 4.3, we added a brief introduction to MediaPipe Pose and a description of the process applied to Dori's physical activity function. We've also added Figures 4-10 to illustrate and demonstrate the pose recognition techniques available in Dori's monitoring service. We think that the pose recognition technology that combines our global optimization method, MPP and the humanoid model is the latest technology.
Additionally, Figure 12 has been added to illustrate Dori's chatbot functionality, and Figure 13 has been added to illustrate the internal processes that run during a conversation between Dori and the user.
Round 2
Reviewer 1 Report
This is a revision of a previously submitted paper that I have reviewed. The authors have satisfied all the concerns I raised in my earlier review. However, there are still several problems that need to be checked by the author.
1. In line 18 of the abstract, the comma is redundant
2. The keyword HCAI is not commonly used, please use the full name.
3. In line 91 of the manuscript, the period is redundant.
4. Figures 5, 8, and 10 are not clear. Please describe the figures in a unified way, especially the representative meaning of the y-axis.
Author Response
The authors deeply appreciate the valuable comments and guidance of the reviewer to improve our paper.
Point 1: In line 18 of the abstract, the comma is redundant
Response 1: The comma is removed.
Point 2: The keyword HCAI is not commonly used, please use the full name.
Response 2: The keyword HCAI is replaced with the full name, human-centered artificial intelligence.
Point 3: In line 91 of the manuscript, the period is redundant.
Response 3: The redundant period is removed.
Point 4: Figures 5, 8, and 10 are not clear. Please describe the figures in a unified way, especially the representative meaning of the y-axis.
Response 4: Figure 5 is different from Figures 8 and 10 in that Figure 5 shows the distance profile of the joint landmark location between the current location and the average location of the previous coordinates, while the other figure shows the identified angular trajectories of representative joints. For a unified description of the three figures, the y-label names are modified.
Reviewer 2 Report
According to the revised manuscript from author, i believe that there are sufficient reasons to accept this paper in its current form and publish it in the journal in due time.Author Response
Point 1: According to the revised manuscript from author, i believe that there are sufficient reasons to accept this paper in its current form and publish it in the journal in due time.
Response 1: The authors deeply appreciate the valuable comments and guidance of the reviewer to improve our paper.